# Changes in Health Facility Readiness for Providing Quality Maternal and Newborn Care After Implementing the Safer Births Bundle of Care Package in Five Regions of Tanzania

**DOI:** 10.3390/healthcare13233060

**Published:** 2025-11-26

**Authors:** Damas Juma, Ketil Stordal, Benjamin Kamala, Dunstan R. Bishanga, Albino Kalolo, Robert Moshiro, Jan Terje Kvaløy, Godfrey Guga, Rachel Manongi

**Affiliations:** 1School of Public Health, KCMC University, Moshi P.O. Box 2240, Tanzania; 2Kigoma Regional Secretariat, Kigoma P.O. Box 125, Tanzania; 3Department of Pediatric Research, Institute of Clinical Medicine, University of Oslo, 03128 Oslo, Norway; 4Department of Research, Haydom Lutheran Hospital, Haydom, Manyara P.O. Box 4000, Tanzania; 5Department of Epidemiology and Biostatistics, Muhimbili University of Health and Allied Sciences (MUHAS), Dar es Salaam P.O. Box 65001, Tanzania; 6Ifakara Health Institute, Dar es Salaam P.O. Box 70373, Tanzania; 7Department of Public Health, St. Francis University College of Health and Allied Sciences, Ifakara P.O.Box 175, Tanzania; 8Department of Paediatrics and Child Health, Muhimbili National Hospital, Dar es Salaam P.O. Box 65000, Tanzania; 9Department of Mathematics and Physics, University of Stavanger, 4036 Stavanger, Norway; 10Department of Research, Stavanger University Hospital, 4068 Stavanger, Norway

**Keywords:** quality improvement, health facility readiness, quality of care, newborn, neonatal care, maternal care, SARA

## Abstract

**Background:** Maternal and newborn morbidity and mortality remain a pressing challenge with uneven progress globally and in Tanzania. The capacity of health facilities to provide quality care is critical to improving outcomes. This study aimed to assess changes in health facilities’ readiness to provide quality maternal and newborn care, and hence aimed to inform improvements in quality-of-care interventions in Tanzania. **Methods:** A before and after assessment of 28 comprehensive emergency obstetric and newborn care health facilities implementing the Safer Births Bundle of Care package in five regions of Tanzania was carried out in December 2020 and January 2023. We adapted the World Health Organization’s Service Availability and Readiness Assessment tool, which covered amenities, equipment, staff, guidelines, medicines, and diagnostic facilities. Composite readiness scores were calculated for each category and results were compared at the health facility level. For categorical variables, we tested for differences by Fisher’s exact test; for readiness scores, differences were tested by linear fixed and mixed model analyses, considering dependencies within the regions. We used *p* < 0.05 as our level of significance and measured change from baseline using a paired t-test. **Results:** The overall readiness improved significantly from 67.6% to 83.7% (*p* < 0.05). Statistically significant improvements were seen in medical equipment (77.1% to 94.0%), diagnostic/treatment commodities (69.3% to 83.1%), and availability of guidelines (50.8% to 96.7%). Changes in amenities (78.1% to 84.2%) and staff (63.0% to 61.7%) were not significant. The overall readiness improved in all facility types and the change was statistically significant in district hospitals and health centres (*p* < 0.05). There were significant differences in improvement between regions (*p* < 0.05) **Conclusions:** The overall readiness has improved significantly, reflecting a positive change. However, there remains a need for further enhancement, particularly in terms of staffing, to ensure high-quality maternal and newborn care. Authorities should take swift action to address the identified gaps, selecting the most effective and practical interventions while closely monitoring progress in readiness and sustaining the gains.

## 1. Introduction

Maternal and neonatal mortality rates remain a pressing global issue. In recognition of this challenge, a worldwide consensus was reached in 2015, setting the Sustainable Development Goals (SDG). SDG 3 set forth to reduce maternal mortality to 70 per 100,000 live births and neonatal mortality to 12 per 1000 live births by 2030 [1]. Progress to reach these goals has been uneven, with some countries making little to no advancement [2]. In 2023, approximately 260,000 women worldwide lost their lives due to pregnancy- and childbirth-related complications, resulting in a maternal mortality rate of roughly 197 per 100,000 live births [3,4]. The global neonatal mortality has declined from 31 in 2000 to 17 per 1000 live births in 2023 [5].

Tanzania, like many developing nations, continues to face higher maternal mortality rates than the global target, despite significant improvements [1,6]. In 2012, the country recorded a maternal mortality rate of 454 per 100,000 live births, which has since declined to 104 per 100,000 [6]. However, neonatal mortality has only slightly improved, decreasing from 26 per 1000 live births in 2010 to 24 per 1000 in 2022, highlighting the need for intensified efforts to meet global benchmarks [6,7].

Despite global efforts to reduce preventable mortality, progress is often hindered by systemic health system challenges such as inadequately skilled personnel and lack of appropriate protocols, necessary infrastructure, essential medicines, and reliable diagnostic tools [2,8,9].

For healthcare to be of high quality, facilities must be sufficiently prepared [8,10]. According to the World Health Organization (WHO), facility readiness refers to the ability of a health centre to provide the services it advertises [8]. Health facility readiness is indispensable not only for achieving positive health outcomes, experience, and trust but also for building health system resilience and achieving universal health coverage [11].

This study examined the readiness of health facilities before and after the 3-year continuous quality improvement (CQI) initiative, the Safer Births Bundle of Care (SBBC). The SBBC integrates scientifically validated clinical and training innovations with new strategies to establish CQI processes and sustain enhanced care, and it commenced in Tanzania in 2020 [12,13,14]. The bundle comprises clinical innovations including the Fetal Heart Rate Monitor (MOYO), newborn heart monitor (Neobeat), and upright bag-mask. Training innovations include a smart training manikin (NeoNatalie Live) for simulating neonatal resuscitation and providing feedback, and Mama Natalie for simulation training in postpartum hemorrhage management. Continuous QI is integrated through regular, on-the-job, low-dose, high-frequency, simulation-based training. Adequate training of local facility champions, who can facilitate CQI simulation training, is considered essential for these processes to occur and stimulate a gradual, sustainable culture change [13,15].

The goal of this study was to evaluate facility readiness to support SBBC implementation and provide insights for other stakeholders seeking to improve the quality of healthcare services.

## 2. Materials and Methods

### 2.1. Study Design and Setting

This before and after study was conducted in 28 health facilities in five selected regions of Tanzania: Manyara, Tabora, Geita, Shinyanga, and Mwanza [12]. The initial facility assessment was performed before the implementation of SBBC and then another assessment was performed three years after the implementation in the same facilities. These health facilities were selected based on the high burden of maternal and perinatal mortality and the volume of deliveries, and absence of other similar interventions. The selected regions account for about 25% of deliveries and about 35% of all maternal and newborn deaths in the country [12]. The annual average number of births that occurred during the study ranged between 400 and 8000 across the study facilities [13]. Two SBBC facilities were excluded from this study because baseline data were not collected, as the decision from the regional and ministry authorities was still pending during data collection, regarding a similar intervention.

The health delivery system in Tanzania is pyramidal, with dispensaries (small outpatient facilities) meant to serve a village at the bottom. Then, there are inpatient health centres, followed by district hospitals (referral points for health centres and dispensaries). Higher up the pyramid, we have regional and then national hospitals. Some health centres and all hospitals are expected to provide comprehensive emergency obstetric and newborn care (CEmONC) services. Health centres typically refer complex cases—or in situations where essential medical personnel or supplies are unavailable—to district hospitals, which may then escalate referrals to regional hospitals. Dispensaries do not provide CEmONC services.

This study was conducted in CEmONC centres where nine cardinal items are expected to be offered: parenteral antibiotics, parenteral anticonvulsants, parenteral oxytocics, assisted delivery, manual removal of the placenta, removal of retained products of conception, basic neonatal resuscitation, cesarian section, and blood transfusion [16].

### 2.2. Data Collection

The data collection for the readiness assessment was performed using a questionnaire adapted from the WHO’s Service Availability and Readiness Assessment (SARA) tool that was initially developed with the participation of several partners to systematically gauge and monitor health services by generating tracer indicators related to the services including emergency obstetric care [17]. We assessed amenities, equipment, staff, guidelines, medicines, and diagnostic facilities. It was administered by trained research teams, composed of medical and nursing health workers with university degrees, to facility and maternity ward in-charges (medical doctors or nurse trained managers). Data collectors were trained for two days at Haydom Research Centre on the data collection tools and data collection process. Data were collected using an interviewer-administered electronic Open Data Kit and then transferred to Excel. Data collection for facility readiness took place first between December 2020 and January 2021 (baseline) and then in December 2023 (endline).

### 2.3. Variables

We assessed 58 items across five domains: seven for amenities, fifteen for basic equipment, eleven for diagnostic and treatment commodities, eleven for staff, and fourteen for guidelines. A 0–1 item score was assigned for each item, according to whether the item was in place or not at a facility. For single item considerations, the percentage of facilities with the item in place was calculated. For each facility, readiness scores for each domain were calculated as the average of the item scores within that domain. The overall readiness was calculated as the average of the readiness scores for each domain, thus giving equal weight to each of the five domains [17].

### 2.4. Data Analysis

The data collected were entered into R software version 4.4.1 for analysis. A comparison of item scores within domains across different levels of facilities was conducted. Fisher’s exact test was used to test for any differences in item scores across different types of health facilities. The comparison between the baseline readiness and endline readiness was performed by comparing the average readiness of each domain and overall readiness at the baseline and endline using a paired *t*-test. For readiness scores, differences were tested by a linear mixed model analysis, considering dependencies within regions. To test for differences in changes between regions, a linear fixed effect model adjusting for facility level was used. The appropriateness of the models was checked by residual plots. We used 0.05 as our level of significance. Due to many tests, for each table with significant sub-scores an indication is given of which tests remain significant after a per-table Bonferroni adjustment for multiple testing.

## 3. Results

The overall readiness in the 28 studied facilities changed from 67.6 percent in 2021 to 83.7 percent in 2023 as shown in Figure 1. The change in overall readiness was statistically significant (*p* < 0.0001), as well as the average scores for commodities, equipment, and guidelines as seen in Table 1. The change in amenities and staffing average score from 63.0 to 61.7 was not statistically significant.

Table 2 tracks the changes in readiness by facility type. The overall readiness improved across all types of facilities. There were increases in most of the domains, except for staff. Observed staff availability decreased in health centres and regional referral hospitals, but the difference was not significant. There was a significant improvement/increase in the overall readiness in both district hospitals and health centres. The change in equipment was significant for all facility levels.

Table 3, Table 4, Table 5, Table 6 and Table 7 show the readiness level at the end of the implementation of the quality improvement initiative in 2023. As shown in Table 3 on the visiting day, facilities had on average 84 percent of items available in the amenities domain. Most items were available in over 90 percent of facilities; however, solar backup was present in only four (14%) of health facilities, all of which were health centres. In the case of a national grid blackout, most facilities rely on generators as their backup power source.

Facilities surveyed had 61.7 percent of the key staff required (Table 4). Staffing was more available in district hospitals and referral hospitals than in health centres. Gynecologists and pediatricians were available in the majority of referral hospitals, while clinical officers and assistant clinical officers were only in health centres and district hospitals.

Availability of equipment was 94 percent on average, with the penguin sucker (a silicon penguin shaped nasal aspirator), newborn suction apparatus, weighing scale, caesarian set, and oxygen cylinders being universally available, as seen in Table 5. The vacuum extractor was the least available equipment, more so in district hospitals and health centres, followed by the room thermometer and small oxygen cylinder for transport.

The average availability of commodities for treatment and diagnosis was 82.1 percent, as seen in Table 6. There was a variation though, with district hospital recording more availability than regional hospitals and health centres. The item that was least available was safe blood.

Table 7 shows the availability of guidelines which was on average 96.7 percent with regional hospitals recording a universal availability.

We also assessed if there would be regional differences in the readiness changes. All regions showed an improvement in readiness and a linear fixed effect regression model showed that there were significant differences between regions in terms of the improvement in readiness. For overall readiness, the mean improvements were in Geita (23.3), Manyara (15.1), Mwanza (14.5), Shinyanga (7.6), and Tabora (19.3), *p* = 0.01. Regions with the lower initial readiness level registered more readiness change. The mean baseline and endline for each region were in Geita 55 to 79, Manyara 71 to 87, Mwanza, 72 to 86, Shinyanga 75 to 83, and Tabora 66 to 85.

## 4. Discussion

This study examined readiness in 28 facilities providing comprehensive emergency obstetric and newborn care in 2023 and compared it with 2021 readiness in the same facilities. The findings indicate positive progress in overall readiness, increasing from 68% to 84%, though there is still room for improvement as is aspired by Tanzania and the global community [1,18,19].

The government of Tanzania has been making efforts to improve health service systems to attain universal health coverage, and this might be at play in the observed readiness improvement in the studied facilities [20]. The change observed in this study is larger than the national readiness change in comprehensive emergency care during a partially overlapping period, which increased from 68% in 2020 to 71% in 2023 [21,22].

The improvement observed might have been, at least in part, attributable to the Plan–Do–Study–Act (PDSA) quality improvement strategy, as facilities in this study were measuring their performance using local data, reviewing and devising action plans, and implementing them for improvement in various areas of readiness [12,13,14,23]. This way of improving readiness and quality of care has been shown to be effective in other studies [24,25,26].

At the endline, the overall readiness was still highest in regional hospitals, followed by district hospitals, and then health centres. This finding aligns with previous studies conducted in Tanzania and other low- and middle-income countries, which have shown that higher-level healthcare facilities tend to exhibit greater overall readiness [18,21,27,28]. This finding highlights a disparity: while CEmONC centres at all levels are expected to provide comparable essential services, mothers and newborns in lower-level facilities often receive lower-quality care. In recent years, however, the Tanzanian government has taken steps to improve existing facilities and construct new ones to expand access to surgical services, particularly cesarean section availability in health centres [20,21]. This indicates that further efforts are needed to strengthen health centres to meet the required readiness standards. Additionally, factors like limited staffing and management capabilities may also play a role in the lower readiness observed in these facilities [29,30].

The improvement in readiness was larger in lower-level facilities than in regional hospitals which had higher initial readiness and therefore less potential to improve further; or, authorities give emphasis to the lesser-performing facilities [22,31,32]. Nevertheless the improvement means that with time, patients in lower-level facilities receive relatively better quality services than is usual, in the context of the usual disparity of poor quality in lower facilities in developing countries [18,21,27,28].

Staffing did not improve significantly. In Tanzania, health employees are usually recruited and work under the central government, local government, or faith-based organizations. However, permits and salaries are funded by the central government. As a result, employment and deployment decisions are not made at the facility level, which helps to explain why there was no improvement in staffing, let alone local issues regarding the rational allocation of staff at the current staffing situation [22]. Additionally, the Tanzanian government has been expanding its healthcare facilities in terms of number and infrastructure, creating a demand for additional staffing—yet the rate of new employment has not kept pace with this growth [20]. Given that the national average for human resource for health remains low, with only 48 percent of the required staff, the insufficient staffing levels at these facilities are unsurprising, sharing trends with other developing nations [33,34].

Guidelines play a crucial role in ensuring that patients receive standardized, evidence-based, or consensus-driven quality care [35]. In this study, guideline availability was nearly universal across all facilities as opposed to the national average availability of guidelines for CEmONC which stood at 56% [22]. The widespread improvement in the availability of various guidelines may be attributed to several factors, including the government and healthcare facilities prioritizing their implementation through dissemination initiatives and distribution. Another reason could be that a long time has passed, allowing for guidelines to integrate into the health system and their adoption, as noted by Pereira and colleagues [36]. For example, in Tanzania, the newborn care guidelines were finalized in 2019, leaving a reasonably long time for their full implementation within the health system [37].

This study has several strengths, including direct facility visits rather than reliance on self-assessment reports. It also encompasses both rural and urban settings and various levels of the health system. However, certain limitations exist: we measured readiness, but the actual service delivery was not observed, and availability does not necessarily indicate utilization. Additionally, staffing levels were not adjusted for client volume or equipment usage. Another constraint is that commodity and amenity availability may fluctuate over time, yet assessments were conducted during a single visit. We did not have control facilities, so we cannot determine whether the changes might have occurred regardless.

## 5. Conclusions

While there has been positive progress, with some areas having an adequate supply of essential items, significant gaps persist in terms of amenities, equipment, diagnostic and treatment commodities, guidelines, and, most notably, staffing. Strengthening health facilities’ readiness is crucial for ensuring high-quality maternal and newborn care. To achieve this, the government, development partners, and other stakeholders must allocate sufficient resources, including amenities, equipment, commodities, and personnel, while carefully selecting the most effective and feasible interventions. We strongly advocate for a continuous quality improvement system to address existing deficiencies. Additionally, regular assessments are vital for monitoring progress in facility readiness and ensuring the successful implementation of measures to enhance service quality. These efforts are essential to reduce preventable maternal and newborn morbidity and mortality.

## Figures and Tables

**Figure 1 healthcare-13-03060-f001:**
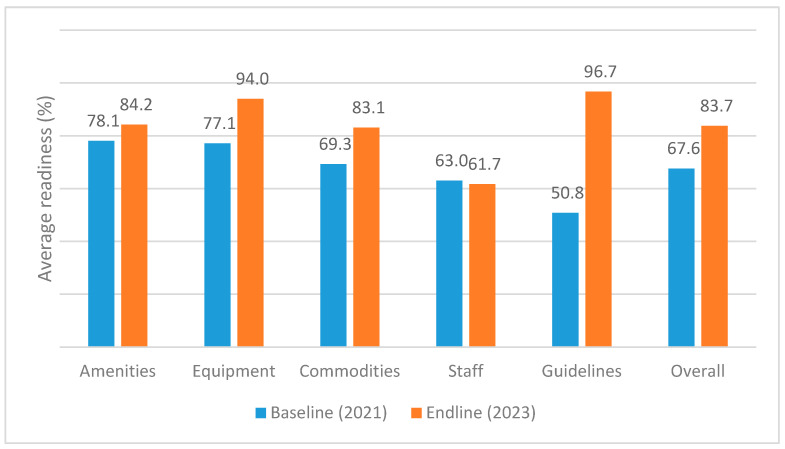
Facility readiness changes in 28 health facilities between 2021 (baseline) and 2023 (endline).

**Table 1 healthcare-13-03060-t001:** Comparison of average readiness before and after the implementation of quality improvement in 28 facilities in Tanzania (baseline 2021 and endline 2023).

Readiness	Baseline N = 28	Endline N = 28	Difference ^1^	95% CI ^1,2^	*p*-Value ^1^
Average Facility Readiness					
Amenity					
Mean (SD)	78.1 (14.3)	84.2 (9.0)	6.1	−0.29, 13	0.056
Staffing					
Mean (SD)	63.0 (16.6)	61.7 (17.5)	−1.3	−8.5, 5.9	0.714
Basic equipment					
Mean (SD)	77.1 (11.5)	94.0 (6.4)	16.9	12.1, 21.7	0.000 ^†^
Commodities					
Mean (SD)	69.3 (19.8)	82.1 (14.1)	12.9	6.3, 19.4	0.000 ^†^
Guidelines					
Mean (SD)	50.8 (20.9)	96.7 (9.2)	45.9	36.5, 55.3	0.000 ^†^
Overall Readiness					
Mean (SD)	67.6 (10.8)	83.7 (6.3)	16.1	10.7, 17.2	0.000 ^†^

^1^ Paired *t*-test; ^2^ CI = confidence interval; ^†^ significant after Bonferroni adjustment.

**Table 2 healthcare-13-03060-t002:** Comparison of average readiness before and after the implementation of quality improvement in 28 facilities in Tanzania by facility type in 28 health facilities.

	Regional Referral Hospital		District Hospital		Health Centre	
Readiness	Baseline, N = 4	Endline, N = 4	Baseline, N = 14	Endline, N = 14	Baseline, N = 10	Endline, N = 10
Amenity						
Mean (SD)	71.4 (11.7)	75.0 (0.0)	81.6 (11.8) *	74.1 (3.3) *	75.7 (17.9)	72.5 (12.9)
Staffing						
Mean (SD)	70.5 (23.9)	65.9 (15.5)	66.2 (12.1)	68.2 (18.4)	55.5 (17.9)	50.9 (12.3)
Basic equipment						
Mean (SD)	85.0 (11.4)	98.3 (3.3)	78.1 (10.6) ***^,^**^†^	96.7 (3.5) ***^,^**^†^	72.7 (11.9) *	88.7 (7.1) *
Commodities						
Mean (SD)	77.5 (18.9)	84.1 (8.7)	69.3 (14.9) ***^,^**^†^	87.0 (8.5) ***^,^**^†^	66.0 (26.3)	74.5 (19.1)
Guidelines						
Mean (SD)	67.9 (14.9) *	100.0 (0.0) *	53.1 (21.8) ***^,^**^†^	95.4 (11.8) ***^,^**^†^	40.7 (17.2) ***^,^**^†^	97.1 (6.9) ***^,^**^†^
Overall Readiness						
Mean (SD)	74.5 (14.6)	84.7 (4.6)	69.7 (5.6) ***^,^**^†^	84.3 (5.6) ***^,^**^†^	62.1 (13.1) ***^,^**^†^	76.8 (5.1) ***^,^**^†^

* Change is statistically significant by paired *t*-test at 0.05 level of significance. ^†^ Significant after Bonferroni adjustment.

**Table 3 healthcare-13-03060-t003:** Endline basic amenities in the 28 facilities providing CEmONC in five regions.

Characteristic	Overall N = 28 ^1^	Regional Referral Hospital N = 4 ^1^	District Hospital N = 14 ^1^	Health Centre N = 10 ^1^	*p*-Value ^2^
	*n* (%)	*n* (%)	*n* (%)	*n* (%)	
Grid electricity	28 (100.0)	4 (100.0)	14 (100.0)	10 (100.0)	
Solar	4 (14.3)	0 (0.0)	0 (0.0)	4 (40.0)	0.02
Generator	27 (96.4)	4 (100.0)	14 (100.0)	9 (90.0)	0.5
Solar or Generator	28 (100.0)	4 (100.0)	14 (100.0)	10 (100.0)	
Water on the visiting day	26 (92.9)	4 (100.0)	13 (92.9)	9 (90.0)	>0.9
Emergency transportation	26 (92.9)	4 (100.0)	14 (100.0)	8 (80.0)	0.2
Fuel for ambulance	26 (92.9)	4 (100.0)	14 (100.0)	8 (80.0%)	0.2
Average readiness score					>0.9
Mean (SD)	84.2 (9.0)	85.7 (0.0)	84.7 (3.8)	82.9 (14.8)	0.89 ^#^

^1^*n* (%); ^2^ Fisher’s exact test; ^#^ Linear mixed model; no tests significant after Bonferroni adjustment.

**Table 4 healthcare-13-03060-t004:** Staff in the 28 health facilities in five regions of Tanzania after implementation.

Characteristic	Overall N = 28 ^1^	Regional Referral Hospital N = 4 ^1^	District Hospital N = 14 ^1^	Health Centre N = 10 ^1^	*p*-Value ^2^
Pediatrician	4 (14.3)	3 (75.0)	1 (7.1)	0 (0.0)	0.005 ^†^
Gynecologist	4 (14.3)	3 (75.0)	1 (7.1)	0 (0.0)	0.005 ^†^
Medical doctor	26 (92.9)	3 (75.0)	14 (100.0)	9 (90.0)	0.12
Assistant Medical Officer ^€^	18 (64.3)	2 (50.0)	11 (78.6)	5 (50.0)	0.3
Clinical Officer ^α^	17 (60.7)	0 (0.0)	10 (71.4)	7 (70.0)	0.040
Assistant Clinical Officer ^¶^	9 (32.1)	0 (0.0)	7 (50.0)	2 (20.0)	0.13
Nurse Officer ^§^	19 (67.9)	4 (100.0)	12 (85.7)	3 (30.0)	0.007
Assistant Nurse Officer ^¥^	27 (96.4)	4 (100.0)	14 (100.0)	9 (90.0)	0.5
Enrolled Nurse ^φ^	28 (100.0)	4 (100.0)	14 (100.0)	10 (100.0)	
Health Information personnel	^¢^ 12 (42.9)	2 (50.0)	8 (57.1)	2 (20.0)	0.2
Anesthetist	26 (92.9)	4 (100.0)	13 (92.9)	9 (90.0)	>0.9
Average Score	61.7	65.9	68.2	50.9	0.089 ^#^

^1^ n (%); ^2^ Fisher’s exact test; ^†^ significant after Bonferroni adjustment; ^#^ linear mixed model; ^§^ holder of a bachelor degree in nursing; ^¥^ holder of a diploma in nursing; ^φ^ holder of a certificate in nursing; ^€^ holder of an advanced diploma in clinical medicine for general medical cores; ^α^ holder of a diploma in clinical medicine to diagnose and treat common medical condition and minor surgeries especially in rural areas; ^¶^ holder of a certificate in clinical medicine—for diagnosis and treatment of medical and minor surgical conditions especially in rural areas; ^¢^ health worker with training in health management information system.

**Table 5 healthcare-13-03060-t005:** Endline availability of equipment in the 28 health facilities in five regions of Tanzania.

Characteristic	Overall N = 28 ^1^	Regional Referral Hospital N = 4 ^1^	District Hospital N = 14 ^1^	Health Centre N = 10 ^1^	*p*-Value ^2^
Weighing scale	28 (100.0)	4 (100.0)	14 (100.0)	10 (100.0)	
Thermometer (room)	25 (89.3)	4 (100.0)	14 (100.0)	7 (70.0)	0.082
Wall clock	25 (89.3)	3 (75.0)	13 (92.9)	9 (90.0)	0.5
Cesarian section set	28 (100.0)	4 (100.0)	14 (100.0)	10 (100.0)	
Vacuum extractor	20 (71.4)	4 (100.0)	9 (64.3)	7 (70.0)	0.5
Resuscitation space next to the delivery bed	27 (96.4)	4 (100.0)	13 (92.9)	10 (100.0)	>0.9
Newborn Suction next to the delivery bed	28 (100.0)	4 (100.0)	14 (100.0)	10 (100.0)	
Newborn Bag/mask next to the delivery bed	28 (100.0)	4 (100.0)	14 (100.0)	10 (100.0)	
Penguin sucker	28 (100.0)	4 (100.0)	14 (100.0)	10 (100.0)	
Mask size 0	27 (96.4)	4 (100.0)	14 (100.0)	9 (90.0)	0.5
Mask size 1	26 (92.9)	4 (100.0)	14 (100.0)	8 (80.0)	0.2
Radiant warmer	26 (92.9)	4 (100.0)	14 (100.0)	8 (80.0)	0.2
Oxygen cylinder/flow metre, humidifier	28 (100.0)	4 (100.0)	14 (100.0)	10 (100.0)	
Small oxygen cylinder for transport	25 (89.3)	4 (100.0)	14 (100.0)	7 (70.0)	0.082
Oxygen concentrator	26 (92.9)	4 (100.0)	14 (100.0)	8 (80.0)	0.2
Average score	94.0	98.3	96.7	88.7	0.0094 ^#^

^1^ n (%); ^2^ Fisher’s exact test; ^#^ linear mixed model.

**Table 6 healthcare-13-03060-t006:** Endline availability of commodities for treatment in the 28 health facilities in five regions.

Characteristic	Overall N = 28 ^1^	Regional Referral Hospital N = 4 ^1^	District Hospital N = 14 ^1^	Health Centre N = 10 ^1^	*p*-Value ^2^
Haemoglobinometer	18 (64.3)	2 (50.0)	10 (71.4)	6 (60.0)	0.7
Glucose metre	23 (82.1)	3 (75.0)	12 (85.7)	8 (80.0)	>0.9
Dextrose 10%	25 (89.3)	4 (100.0)	13 (92.9)	8 (80.0)	0.7
Normal saline	28 (100.0)	4 (100.0)	14 (100.0)	10 (100.0)	
Vitamin K	28 (100.0)	4 (100.0)	14 (100.0)	10 (100.0)	
Ampicillin/gentamicin	26 (92.9)	4 (100.0)	14 (100.0)	8 (80.0)	0.2
Adrenaline injection	25 (89.3)	4 (100.0)	14 (100.0)	7 (70.0)	0.082
Phenobarbital injection	21 (75.0)	4 (100.0)	12 (85.7)	5 (50.0)	0.11
Magnesium Sulphate	28 (100.0)	4 (100.0)	14 (100.0)	10 (100.0)	
Safe blood 24 hrs	28 (100.0)	4 (100.0)	14 (100.0)	10 (100.0)	
Blood out of stock in 90 days before the study	3 (10.7)	0 (0.0)	3 (21.4)	0 (0.0)	0.4
Average score	82.1	84.1	87.0	74.5	0.082 ^#^

^1^ *n* (%); ^2^ Fisher’s exact test; ^#^ linear mixed model.

**Table 7 healthcare-13-03060-t007:** Endline availability of guidelines in the 28 health facilities in five regions of Tanzania.

Characteristic	Overall N = 28 ^1^	Regional Referral Hospital N = 4 ^1^	District Hospital N = 14 ^1^	Health Centre N = 10 ^1^	*p*-Value ^2^
National neonatal guidelines	28 (100.0)	4 (100.0)	14 (100.0)	10 (100.0)	
Essential newborn care guidelines	27 (96.4)	4 (100.0)	13 (92.9)	10 (100.0)	>0.9
Standard operating procedure set	26 (92.9)	4 (100.0)	12 (85.7)	10 (100.0)	0.6
Newborn triage checklist	25 (89.3)	4 (100.0)	12 (85.7)	9 (90.0)	>0.9
Discharge feedback form	25 (89.3)	4 (100.0)	13 (92.9)	8 (80.0)	0.7
Referral form	27 (96.4)	4 (100.0)	14 (100.0)	9 (90.0)	0.5
HBB poster	28 (100.0)	4 (100.0)	14 (100.0)	10 (100.0)	
Prolonged labour	28 (100.0)	4 (100.0)	14 (100.0)	10 (100.0)	
Pre-eclampsia	28 (100.0)	4 (100.0)	14 (100.0)	10 (100.0)	
Antepartum hemorrhage	27 (96.4)	4 (100.0)	13 (92.9)	10 (100.0)	>0.9
Postpartum hemorrhage	28 (100.0)	4 (100.0)	14 (100.0)	10 (100.0)	
Abnormal fetal heart rate	27 (96.4)	4 (100.0)	13 (92.9)	10 (100.0)	>0.9
PPROM	27 (96.4)	4 (100.0)	13 (92.9)	10 (100.0)	>0.9
Antenatal Corticosteroid	28 (100.0)	4 (100.0)	14 (100.0)	10 (100.0)	
Average readiness	96.7	100.0	95.4	97.1	0.6069 ^#^

^1^ n (%); ^2^ Fisher’s exact test; ^#^ linear mixed model.

## Data Availability

Data are available on reasonable request from Haydom Lutheran Hospital, P.O. Box 9000 Haydom, Manyara, Tanzania Tel. + 255(0)27 253 3194/5 Fax + 255(0)27 253 3734 E-mail: post@haydom.co.tz. This is due to ethical and privacy reasons.

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
