# Peer review of "Changes in Health Facility Readiness for Providing Quality Maternal and Newborn Care After Implementing the Safer Births Bundle of Care Package in Five Regions of Tanzania"

_healthcare, 2025, doi:10.3390/healthcare13233060_

Round 1
Reviewer 1 Report
Comments and Suggestions for Authors
Thank you for submitting this interesting paper. I have provided a detailed document suggesting how your work might be improved. Remember that you are sending this to an international audience but currently it is very SSA centric. Some terms need further explanation to really demonstrate to readers what your study has explored.

Reviewer 2 Report
Comments and Suggestions for Authors
Childbirth is a traumatic process and associated with significant mortality for both the mother and the young one, it is also an important measure of healthcare in different geographies. Over the past decades, mother and new born mortality has come down in most parts of the world, but it remains a concern in many countries. Research projects that aim to reduce mortality, by identifying and assessing key determinants for safe child birth are of national importance. The authors must be commended for taking up such a study, in a developing country where this mortality is still high. The paper has been well written and the results presented in a way that is easy to analyse and understand. Assessment of 28 Comprehensive Emergency Obstetric and New born Care health facilities, helps quantitate the implementation of Safer Births Bundle of Care. The tool used is the WHO’s Service Availability and Readiness Assessment tool, which is a tool validated in many countries.
This study has demonstrated a statistical improvement in 5 of the 6 determinants of facility readiness between 2021 and 2023. It is heartening to note that this improvement is not restricted to Regional Reference Hospitals, but also across District Hospitals and Health Centres. This rather important, since in many under developed and developing countries large referral hospitals continue to improve, but in the interiors, there is no change. Since a larger percentage of population lives in the interiors, improvement needs to be spread evenly across the country.
The authors have documented changes across two calendar years, however during one of the periods of data collection (Dec 2020 - Jan 2021) the SARS-COV -2 virus pandemic was raging across Tanzania. The pandemic is probably the first global pandemic experienced by our generation and it affected medical services, all over the world. It is not definitely known to what extent the pandemic affected facility readiness for mothers and their new born in Tanzania. One wonders whether the growth and improvement seen was real, or it was due to altered medical facilities during the pandemic. Secondly the authors have not collected data on maternal and new born mortality in the 28 centres that they studied. Had this data been available, one could have checked the correlation between the change in facilities with the change in mortality, if any.
Reviewer 3 Report
Comments and Suggestions for Authors
This study has weakened the findings as the methodology is very modest. Improving the analysis should be done.

Reviewer 4 Report
Comments and Suggestions for Authors
The manuscript reports a before-and-after assessment of 28 health facilities implementing the Safer Births Bundle of Care (SBBC) across five Tanzanian regions. Using the WHO Service Availability and Readiness Assessment (SARA) framework, it compares readiness in 2020–2021 and 2023. The study finds a statistically significant improvement in overall readiness—from 67.6% to 83.7%—driven by gains in equipment, commodities, and guideline availability, though staffing levels remained static. However, for a high-impact journal, the manuscript is too descriptive, methodologically simplistic, and analytically limited to warrant publication.
- The study design is described as “before and after,” but lacks a comparison group (e.g., facilities not exposed to the SBBC). Without a counterfactual, the reported improvements could reflect secular trends, national investments, or concurrent initiatives.
- The timing of assessments (2020–2021 vs. 2023) coincides with major COVID-19–related disruptions and subsequent health-system recovery efforts, which may independently explain some improvements.
- No adjustment for multiple testing was performed despite numerous domain-specific comparisons.
The analysis remains purely descriptive. For publication in a high-impact journal, readers expect:
Multivariate modeling of facility characteristics associated with higher readiness gains.
Exploration of regional variation and factors driving differential improvement.
Triangulation with SBBC implementation intensity or fidelity metrics (e.g., training frequency, supervision visits).
The more you can look at, the better for the article.
- The authors claim coverage of “25% of all deliveries and 35% of maternal deaths,” but this requires clarification and citation—are these regional aggregates or study-facility statistics?
- The study should disaggregate staff categories by cadre, compare against WHO density benchmarks, and discuss implications for functional readiness (having equipment is meaningless without trained personnel).
- Statements such as “the improvement might have been partly due to the Plan–Do–Study–Act strategy” are speculative; no data were collected to substantiate this. The authors must temper these claims or include qualitative or process data showing facility-level QI cycles.
- English is clear but repetitive (“readiness improved… readiness improved…”). Substantive editing for conciseness is needed. Avoid first-person phrasing (“We still see at the endline…”).
Round 2
Reviewer 1 Report
Comments and Suggestions for Authors
Thank you for resubmitting your paper. You have obviously worked hard to strengthen it. There are a few things that I recommend that will provide more clarity: -
- More detail is required about the roles of professionals. What is, for example, a clinical officer. This is a unique role to SSA countries. A table with a description of each role would be appreciated.
- . A minor point - Page 13, line 325. Delete the word 'enough' to make the sentence read fluidly.
Reviewer 3 Report
Comments and Suggestions for Authors
Authors have responded and revised as the reviewer suggested. We accept this revised manuscript version.
Reviewer 4 Report
Comments and Suggestions for Authors
The authors have submitted a substantially revised version of their manuscript titled “Changes in health facility readiness for providing quality maternal and newborn care after implementing the Safer Births Bundle of Care package in five regions of Tanzania.” The revision demonstrates clear improvements in structure, analytical detail, and methodological transparency. The authors have addressed most of the previous comments thoughtfully, adding regional fixed-effects regression analyses, Bonferroni adjustments for multiple testing, and clarifying key elements such as study coverage and COVID-19 context. The manuscript is now more coherent and statistically sound. However, despite these commendable revisions, several conceptual and methodological limitations remain that continue to constrain the scientific strength and novelty of the paper. The work remains predominantly descriptive, and the absence of a counterfactual or control group continues to limit causal inference.
- The authors argue that COVID-19 had minimal influence on Tanzanian healthcare operations, citing one publication. However, given the global disruption in supply chains and workforce deployment, a stronger justification or sensitivity analysis is warranted to support this claim.
- Some sentences still imply causation (“the improvement might have been partly due to the PDSA strategy”) without supportive evidence. These should be reframed as hypotheses or possibilities. No process indicators or qualitative data are provided to substantiate the role of PDSA cycles or SBBC fidelity.
- The English has improved considerably, though some repetitive phrasing remains (“readiness improved…”). Further stylistic editing for conciseness is recommended.
- Ensure all acronyms (SBBC, CEmONC, PDSA) are defined at first use in the abstract and text.
